# Potential and Alternative Bioactive Compounds from Brown *Agaricus bisporus* Mushroom Extracts for Xerosis Treatment

**Nichcha Nitthikan** [1], **Pimporn Leelapornpisid** [2], **Ornchuma Naksuriya** [2], **Nutjeera Intasai** [3] and **Kanokwan Kiattisin** [2,4],*

1. PhD Degree Program in Pharmacy, Faculty of Pharmacy, Chiang Mai University, Chiang Mai 50200, Thailand
2. Department of Pharmaceutical Sciences, Faculty of Pharmacy, Chiang Mai University, Chiang Mai 50200, Thailand
3. Division of Clinical Microscopy, Department of Medical Technology, Faculty of Associated Medical Sciences, Chiang Mai University, Chiang Mai 50200, Thailand
4. Innovation Center for Holistic Health, Nutraceuticals and Cosmeceuticals, Faculty of Pharmacy, Chiang Mai University, Chiang Mai 50200, Thailand
* Correspondence: kanokwan.k@cmu.ac.th

**Abstract:** This study aimed to investigate the ability of brown *Agaricus bisporus* extracts to enhance xerosis treatment via their biological activities, including their antioxidant, anti-aging, and anti-inflammation. Brown *A. bisporus* ethanol extract (EE) and brown *A. bisporus* water extract (WE) contained ergothioneine and gallic acid as their major compounds, as detected by HPLC, respectively. The WE exhibited the highest total polysaccharide content ($734.04 \pm 0.03$ mg glucose/g extract) and total phenolic content ($190.90 \pm 0.07$ mg gallic acid/g extract). The WE exhibited an inhibitory effect of $83.34 \pm 18.66\%$ on a collagenase enzyme, whereas the EE inhibited the elastase enzymes by $81.26 \pm 4.37\%$. In addition, the EE also demonstrated strong activities against DPPH, with an $IC_{50}$ $0.30 \pm 0.04$ mg/mL, ABTS with a TEAC value of $8.06 \pm 0.08$ μM Trolox/g extract, and a FRAP assay with a FRAP value of $390.50 \pm 0.32$ mM $FeSO_4$/g. In addition, all extracts were non-cytotoxic and could decrease the secretion of IL-6 and TNF-α in HaCaT cells. Therefore, brown *A. bisporus* extracts might be a potential natural raw material that can be further used in cosmeceutical products for xerosis treatment due to their good efficacy.

**Keywords:** brown *Agaricus bisporus*; xerosis; anti-inflammation; anti-aging; skin barrier function

## 1. Introduction

The unwelcome visible signs of skin aging are skin inflammation and dryness. Xerosis is extremely dry skin in people older than 60. Chronic dry skin can lead to severe xerosis characterized by skin inflammation, pruritus, and fissured and cracked skin [1]. As the same time, rashes and pruritus are signs of xerosis and are caused by a lack of epidermal lipids and skin changes in the stratum corneum (SC), especially during the keratinization process. The SC consists of dead epidermal cells and corneocytes surrounded by a multi-lamellar lipid matrix (40–50% ceramides, 20–25% cholesterol, and 10–15% fatty acids). With aging, the skin structure becomes thinner, causing water, moisture, collagen, and elastin loss. Moisturizing substances are known as the natural moisturizing factors (NMFs) in the skin barrier and consist of amino acids and their derivatives, pyrrolidone carboxylic acid, urocanic acid, urea, glycerol, and lactic acid. These moisturizing substances act as humectants that are essential to the permeability barrier and water content. A deficiency in skin barrier function, such as a decrease in ceramides, leads to a deficiency in skin hydration in the epidermis and an increase in transepidermal water loss (TEWL), which is associated with itching and inflammation. In addition, keratinocytes are involved in skin barrier functions against pathogen entry into the skin and are responsible for immune responses [2]. Keratinocytes are stimulated by triggering factors that induce skin

inflammation by releasing pro-inflammatory cytokines, such as interleukin (IL)-6, IL-8, IL-2, and tumor necrosis factor (TNF)-$\alpha$. Among these cytokines, TNF-$\alpha$ is involved in the promotion of inflammatory reactions through the activation of the cytokine IL-6 [3]. Increased IL-6 is associated with inflammatory processes that lead to chronic skin itching and skin inflammation in many skin diseases, including xerosis, atopic dermatitis, and psoriasis [4].

Optimal topical skin care for xerosis treatment contains remoisturizing agents, such as urea and glycerol, which serve as humectants and aid in lipid replenishing [5]. It is vital to manage xerosis by maintaining epidermal barrier functions and by protecting tissues and skin structures from infection and physical damage. Moreover, combinational treatments involving antibiotics, anti-histamines, and corticosteroids can treat skin inflammation, repair altered skin barrier function, and help to reduce itching [6]. New treatment approaches are being investigated with various plants and fungi that contain many bioactive compounds that have antioxidant, anti-inflammatory, anti-aging, and moisturizing effects. Thus, natural active ingredients present a potential alternative treatment pathway for xerosis. Previous research has revealed that mushroom species have been used for medicinal purposes ranging from the treatment of skin diseases to the prevention of other diseases, such as hypertension, diabetes, hypercholesterolemia, and cancer, due to their bioactive compounds related to antioxidant, anti-inflammatory, and antitumor activities [7].

Brown *Agaricus bisporus* (brown *A. bisporus*) is commonly known as the brown button mushroom or portobello mushroom. Brown *A. bisporus* is a highly recommended edible mushroom that can improve health due to its nutritional properties, proteins, vitamins, and minerals [8]. It displays several biological activities related to skin whitening and skin moisturizing as well as antioxidant, anti-aging, and anti-inflammatory effects, and offers a new perspective for xerosis treatment. Moreover, there have been no reports on the use of brown *A. bisporus* bioactivities as cosmeceuticals to achieve anti-inflammatory effects or to improve skin barrier function improvement.

Therefore, this research aimed to investigate the effect of brown *A. bisporus* extracts on biological activities, antioxidant, anti-aging, and anti-inflammatory activities, for xerosis treatment.

## 2. Materials and Methods

### 2.1. Materials

Dulbecco's modified Eagle's medium (DMEM), fetal bovine serum (FBS), and 3-(4,5-dimethylthiazol-2-yl)-2,5-diphenyltetrazolium bromide (MTT) were purchased from Gibco, Thailand. Penicillin–streptomycin and trypan blue were purchased from Gibco/Invitrogen (Carlsbad, CA, USA). Human immortalized keratinocyte (HaCaT) cells were purchased from Pacific Science Co., Ltd., Bangkok, Thailand (Catalog No. EP-CL-0090, Lot No. 8300I142008). Human IL-6 and TNF-$\alpha$ were purchased from Elabscience Biotechnology, Houston, TX, USA. 2,2′-azino-bis (3-ethylbenzthiazoline-6-sulphonic acid) (ABTS), 2,2-Diphenyl-1-picrylhydrazyl (DPPH), D-glucose, *N*-Succinyl-Ala-Ala-Ala-p-nitroanilide (AAAVPN), *N*-[3-(2-Furyl)acryloyl]-Leu-Gly-Pro-Ala (FALGPA), hyaluronidase enzyme, elastase enzyme, collagenase enzyme, bovine serum albumin (BSA), epigallocatechin 3 gallate, gallic acid, ergothioneine, trolox, sodium chloride (NaCl), sodium acetate (CH$_3$COONa), ferrous chloride (FeCl$_2$), ferric chloride (FeCl$_3$), ferrous sulfate (FeSO$_4$) and lipopolysaccharide (LPS) were purchased from Sigma–Aldrich, Steinheim Germany. Dimethyl sulfoxide (DMSO), analytical grade 95% ethanol, acetic acid, hydrochloric acid (HCl), sulfuric acid, formic acid, acetonitrile, triethylamine, calcium chloride (CaCl$_2$), tricine, potassium persulfate (K$_2$S$_2$O$_8$), sodium dihydrogen phosphate (NaH$_2$PO$_4$), disodium phosphate (Na$_2$HPO$_4$), sodium phosphate (Na$_3$PO$_4$), and aluminum chloride (AlCl$_3$) were purchased from Labscan, Dublin, Ireland. 2,4,6-Tris(2-pyridyl)-s-triazine (TPTZ) was purchased from FlukaBuchs, Buchs, Switzerland. Anthrone was purchased from Himedia®, Mumbai, India. Tannic acid was purchased from LobaChemie, India. FolinCiocalteu reagent was purchased from Supelco®, Bellefonte, PN, USA.

### 2.2. Brown A. bisporus Preparation

In this study, brown *A. bisporus* was freshly harvested from the royal project in Thailand between 2020 and 2021. The mushroom's fruiting body, consisting of a cap, gills, tubes, and stipe, was collected and some residuals were removed using water. Brown *A. bisporus* was dried at $50 \pm 2$ °C in a hot air oven (Universal oven Memmert UN 55, Schwabach, Germany) for an hour and ground into powder with a blender (600 W, Viva Collection Blender Phillip, Bangkok, Thailand). The sample was kept in an amber glass bottle at room temperature for extraction.

### 2.3. Brown A. bisporus Extraction

The extraction process was performed following a previous protocol with some modifications [9]. The mushroom powder (100 g) was extracted with 500 mL of 95%v/v ethanol at room temperature for 48 h and repeated in triplicate. The mixture was filtered through a Whatman® filter paper No.1 (Merck KgaA, Darmstadt, Germany) and then concentrated using a rotary evaporator (Eyela, Tokyo, Japan) at 50 °C. The ethanol extract of brown *A. bisporus* (EE) was kept in an amber glass bottle at $2 \pm 2$ °C. The remaining residue powder was dried in a hot air oven at $50 \pm 2$ °C for 1 h to remove the ethanol. For the aqueous extraction, the residue was then re-extracted with hot water at 85 °C for 4 h. The mixture was filtered through a Whatman® filter paper No.1 and the water was removed using a rotary evaporator at $50 \pm 2$ °C. The water extract from brown *A. bisporus* (WE) was kept in an amber glass bottle at $2 \pm 2$ °C for all experiments.

### 2.4. Measurement of Total Polysaccharides Content, Total Phenolics Content, and Total Flavonoids Content

#### 2.4.1. Determination of Total Polysaccharides Content

The total polysaccharide content of each extract was evaluated by an anthrone assay with some modifications [10]. The anthrone reagent comprises 0.2%w/v anthrone in sulfuric acid. Briefly, 50 μL of a sample at a concentration of 10 mg/mL was mixed with 100 μL of anthrone reagent. The mixture was incubated at 80 °C in a water bath (Memmert Waterbath WNB, Schwabach, Germany) for 5 min and its absorbance was measured at 625 nm using a UV-vis spectrophotometer (Shimadzu, Japan). The analytical curve was plotted using standard D-glucose with different concentrations (y = 3.9027x + 0.4686), where y is the absorbance value and x is the standard glucose content (mg). The results are expressed as milligrams of glucose equivalent per gram of extract using the following equation:

$$\text{Total polysaccharides content (mg glucose/g extract)} = (c \times V \times D)/N \qquad (1)$$

where c is the concentration of glucose (mg), V is the sample volume (mL), D is the dilution factor, and N is the weight of the sample (g).

#### 2.4.2. Determination of Total Phenolics Content

The total phenolics content of each extract was determined by a Folin–Ciocalteu assay with some modifications [11]. Briefly, 10 μL of a sample at a concentration of 10 mg/mL was mixed with 100 μL of Folin–Ciocalteu reagent (1:9; Folin–Ciocalteu reagent:distilled water). We added 7.5%w/v sodium carbonate solution (90 μL) to the mixture. After 30 min, the mixture was kept in the dark and its absorbance was measured at 765 nm using a microplate reader (SPECTROstar Nano, Aylesbury, UK). The analytical curve was plotted using gallic acid with different concentrations (y = 2.4791x − 0.0558), where y is the absorbance value and x is the content of gallic acid (mg). The results are expressed as milligram gallic acid equivalent (GAE) per gram of extract using the following equation:

$$\text{Total phenolics content (mg GAE/g extract)} = (c \times V \times D)/N \qquad (2)$$

where c is the concentration of gallic acid (mg), V is the sample volume (mL), D is the dilution factor, and N is the weight of the sample (g).

### 2.4.3. Determination of Total Flavonoids Content

The total flavonoids content of each sample was determined by an aluminum chloride colorimetric assay with some modifications [11]. Briefly, 50 μL of a sample was mixed with 10 μL of 10%w/v aluminum chloride solution followed by 180 μL of ethanol in a 96-well plate. Then, 10 μL of 1 M sodium acetate was added to the mixture and kept at room temperature in the dark for 40 min. The absorbance was measured at 415 nm using a microplate reader (SPECTROstar Nano, Aylesbury, UK). The analytical curve was plotted using quercetin with different concentrations (y = 2.4026 + 0.4801), where y is the absorbance value, and x is the content of quercetin (mg). The results are expressed as milligram quercetin equivalent (QE) per gram of extract using the following equation:

$$\text{Total flavonoids content (mg QE/g extract)} = (c \times V \times D)/N \qquad (3)$$

where c is the concentration of quercetin (mg), V is the sample volume (mL), D is the dilution factor, and N is the weight of the sample (g).

### 2.5. Determination of Chemical Components by High Performance Liquid Chromatography (HPLC)

Based on the literature reviews, amino acids and phenolic acids are an active substance found in *A. bisporus*. Therefore, ergothioneine and gallic acid were chosen as chemical markers in this study. The marker compounds presented in the extracts were analyzed using HPLC. Each sample (50 mg/L) was dissolved in methanol and filtered through a 0.45 μM polyvinylidene difluoride filter before injection into HPLC. Ergothioneine in the extracts was analyzed following a previous HPLC protocol with some modifications [12]. It was detected using an Agilent HPLC 1100 (Agilent Technologies, Waldbronn, Germany) with a C8 column (Eclipse XDB-Phenyl 4.6 mm ID × 250 mm, 5 μM) as a stationary phase. The isocratic mobile phase was (A) 3%*v/v* acetonitrile, 0.05%*v/v* triethylamine (pH 7.5), and 50 mM sodium phosphate; (B) 99.99%v/v acetonitrile; and (C) distilled water with a ratio of 5:15:80. The flow rate was 0.5 mL/min and the injection volume was 10 μL. The ergothioneine standard solutions were prepared for the standard curve at concentration from 25 to 125 mg/L at a wavelength of 254 nm.

Gallic acid was used as the phenolic standard. The mobile phase was (A) 0.1%*v/v* formic acid in distilled water and (B) 0.1%*v/v* in acetonitrile with a ratio of 85:15, a flow rate of 0.5 mL/min, and an injection volume of 10 μL. A C8 column (Eclipse XDB-Phenyl 4.6 mm ID × 250 mm, 5 μM) was used as the stationary phase. A gallic acid standard solution was prepared for the standard curve at concentrations from 10 to 100 mg/L at a wavelength of 280 nm [13].

### 2.6. Determination of Antioxidant Activity

### 2.6.1. DPPH Radical Scavenging Assay

The free radical scavenging activity of extracts was estimated by DPPH assay with some modifications [14]. Each sample was prepared in the range of 3.13–50 mg/mL. Firstly, 20 μL of the sample was mixed with 180 μL of 120 mM DPPH in ethanol. The mixture was incubated at room temperature and protected from light for 30 min. The absorbance was measured at 520 nm using a microplate reader (SPECTROstar Nano, Aylesbury, UK). The control was served as the DPPH solution without adding a sample. Ergothioneine, gallic acid, and ascorbic acid were used as the standard antioxidants. The half maximal inhibitory concentration ($IC_{50}$) was calculated from the plotted linear graph of sample concentrations versus percentage of inhibition.

### 2.6.2. Ferric-Reducing Antioxidant Power (FRAP) Assay

The ferric-reducing antioxidant power of each extract was measured by a FRAP assay with some modifications [14]. The FRAP reagent comprises 300 mM acetate buffer, 10 mM TPTZ dissolved in 40 mM of 37%*v/v* hydrochloric acid, and 20 mM ferric chloride solution with the ratio of 50:5:5. Briefly, 20 μL of the sample at a concentration of 1 mg/mL was

mixed with 180 µL of FRAP reagent. The mixture was incubated at room temperature for 5 min. The dark blue solution was measured at 595 nm using a microplate reader (SPECTROstar Nano, Aylesbury, UK). Ergothioneine, gallic acid, and ascorbic acid were used as the standard antioxidants. The analytical curve was plotted using ferrous sulfate with different concentrations as a standard curve (y = 2.6033x + 0.3179), where y is the absorbance value and x is the content of standard ferrous sulfate (mg). The results are expressed as mg of ferrous sulfate equivalent per gram of sample or a FRAP value using the following equation:

$$\text{FRAP value (mg FeSO}_4/\text{g of extract)} = (c \times V \times D)/N \tag{4}$$

where c is the concentration of ferrous sulfate (mg), V is the sample volume (mL), D is the dilution factor, and N is the weight of the sample (g).

### 2.6.3. ABTS Radical Scavenging Assay

The ABTS radical scavenging activity of each extract was determined by an ABTS assay with some modifications [15]. The ABTS solution was prepared by blending 2 mL of 7 mM ABTS stock solution with 35.50 µL of 2.45 mM potassium persulfate stock solution and kept in the dark at room temperature for 16 h. ABTS solution was then diluted with deionized water at a ratio of 1:120 to obtain ABTS working solution. Briefly, 10 µL of the sample and 240 µL of ABTS solution were mixed and protected from light. After 6 min, the reaction was measured at 734 nm using a microplate reader (SPECTROstar Nano, Aylesbury, UK). Ergothioneine, gallic acid, and ascorbic acid were used as the standard antioxidants. The analytical curve was plotted using trolox with different concentrations as a standard curve (y = 0.0019x + 0.4525), where y is the absorbance value and x is the content of standard trolox (µM). The results are expressed as the trolox equivalent antioxidant capacity (TEAC) according to the equation:

$$\text{TEAC} = IC_{50} \text{ of trolox } (\mu M/L)/IC_{50} \text{ of sample } (\mu M/L) \tag{5}$$

### 2.7. Determination of Anti-Aging Activities

### 2.7.1. Hyaluronidase Inhibitory Assay

The extracts were tested for the inhibition of hyaluronidase enzyme by the turbidimetric method with some modifications [16]. The hyaluronidase enzyme activity accounted for more than 90% of enzyme activity in this experiment. We prepared 0.3 M phosphate buffer (PBS) at pH 5.35 and used it as the buffer solution in this assay. Firstly, each 100 µL of 2 mg/mL of hyaluronidase enzyme from bovine testis (E.C. 3.2.1.3.5) in PBS was incubated with 50 µL of the sample (1 mg/mL) in a test tube at 37.5 °C in a water bath (Memmert Waterbath WNB, Schwabach, Germany) for 10 min. After that, 100 µL of 0.03% *w/v* hyaluronic acid in PBS was added and then incubated at 37.5 °C. After 45 min, 1 mL of acetic bovine serum albumin, consisting of sodium acetate, acetic acid, and bovine serum albumin (pH 3.75), was added to precipitate the undigested hyaluronic acid. The mixture was kept at room temperature for 10 min and then the absorbance was measured at 600 nm using a microplate reader (SPECTROstar Nano, Aylesbury, UK). Ergothioneine, gallic acid, and tannic acid were used as the reference standards. The absorbance value in the absence of enzymes was used as a control. The percent inhibition of the hyaluronidase enzyme is calculated by the following equation:

$$\% \text{ Hyaluronidase inhibition} = (A_{\text{sample}}/A_{\text{control}}) \times 100 \tag{6}$$

where $A_{\text{sample}}$ is the absorbance of the sample, hyaluronidase enzyme solution, hyaluronic acid solution, and acetic albumin acid solution. $A_{\text{control}}$ is the absorbance of deionized water, hyaluronic acid solution, and acetic albumin acid solution.

2.7.2. Collagenase Inhibitory Assay

The collagenase inhibition was evaluated following the method of Thring et al. with some modifications [17]. The collagenase enzyme activity should be more than 90% before the testing experiment. The assay was performed in 50 mM Tricine buffer at pH 7.5 with 400 mM sodium chloride and 10 mM calcium chloride. Collagenase from Clostridium histolyticum (ChC—EC .3.3.23.3) was dissolved in a Tricine buffer at a concentration of 1 mg/mL. The substrate N-[3-(2-Furyl)acryloyl]-Leu-Gly-Pro-Ala (FALGPA) was performed in 2 mM in a tricine buffer. Briefly, 10 μL of the sample with a concentration of 1 mg/mL was incubated with 40 μL of collagenase enzyme for 15 min, then 50 μL of FALGPA was added to the sample. After adding the substrate, the absorbance was immediately measured at 340 nm with kinetic mode using a microplate reader (SPECTROstar Nano, Aylesbury, UK). Ergothioneine, gallic acid and epigallocatechin gallate were used as reference standards. The percentage inhibition of collagenase is calculated by the following equation:

$$\% \text{ Collagenase inhibition} = (A_{control} - A_{sample})/(A_{control}) \times 100 \tag{7}$$

where $A_{control}$ is the absorbance of the reaction of deionized water, collagenase enzyme solution, and substrate. $A_{sample}$ is the absorbance of the reaction of sample solution, collagenase enzyme solution, and substrate.

2.7.3. Elastase Inhibitory Assay

The elastase inhibition of the extracts was performed by measuring the product during the reaction of the elastase enzyme and substrate using a spectrophotometric method with some modifications [17]. The elastase enzyme activity should be more than 90% before the testing experiment. Briefly, 100 mM of tris with HCl was prepared and used as a tris–HCl buffer (pH 8). The elastase enzyme from a porcine pancreas at a concentration of 2 mg/mL was dissolved in the tris–HCl buffer. The substrate as N-Succinyl-Ala-Ala-Ala-p-nitroanilide (AAAVPN) was prepared at a concentration of 4.4 mM in a tris–HCl buffer. Briefly, 50 μL of the sample at a concentration of 1 mg/mL was mixed with 25 μL of substrate then incubated for 20 min at room temperature. After that, 25 μL of the enzyme was added into the mixture to start the reaction. The absorbance was immediately measured at 410 nm with kinetic mode using a microplate reader (SPECTROstar Nano, Aylesbury, UK). Ergothioneine, gallic acid, and epigallocatechin gallate were used as the reference standards. The percentage inhibition of the elastase enzyme was calculated using the following equation:

$$\% \text{ Elastase inhibition} = (A_{control} - A_{sample})/(A_{control}) \times 100 \tag{8}$$

where $A_{control}$ is the absorbance of the deionized water, elastase enzyme solution, and substrate. $A_{sample}$ is the absorbance of the sample, elastase enzyme solution, and substrate.

*2.8. Cell Culture*

2.8.1. Determination of Cytotoxicity

The cytotoxicity of the extracts on HaCaT cells was evaluated by MTT assay [18]. In brief, HaCaT cells were seeded at $1 \times 10^4$ cells/well in 96-well plates and incubated at 37 °C with 5% $CO_2$ for 24 h. Then, the media were replaced by a sample solution in a concentration range from 37.5 to 300 μg/mL and incubated at 37 °C under 5% $CO_2$. After 48 h of incubation, 15 μL of the MTT reagent (5 mg/mL) was added and incubated for 4 h. The MTT solution was removed and 200 μL of DMSO was added to solubilize the purple formazan crystal. The solubilized formazan was measured using a microplate reader at 570 nm with a reference wavelength of 630 nm. The experiments were conducted in triplicate. The percentage of cell viability was calculated by the following equation:

$$\% \text{ Cell viability} = (\text{OD of sample well}/\text{OD of vehicle control}) \times 100 \tag{9}$$

where the OD of the sample is the absorbance of the treated cells. The OD of the vehicle control is the absorbance of the untreated cells.

### 2.8.2. Quantification of IL-6 and TNF-α Secretion by ELISA

HaCaT cells were used to determine the anti-inflammatory cytokines secretion (IL-6 and TNF-α) in culture supernatant. In brief, HaCaT cells were seeded in 96-well plates with $1 \times 10^4$ cells/well, after which the cells were incubated at 37 °C with 5% $CO_2$ for 24 h. HaCaT cells were pre-treated with a non-toxic concentration of the sample for 2 h. The medium was removed and then LPS (1 μg/mL) was added to each well and incubated in a $CO_2$ incubator for 24 h. The supernatant was collected and centrifuged at $10,000 \times g$ for 5 min. The concentration of cytokines for each sample was detected using a sandwich enzyme-linked immunosorbent assay (ELISA) kits (Elabscience, Houston, TX, USA) following the manufacturer's instructions.

### 2.9. Statistical Analysis

All results are expressed as the mean $\pm$ SD. The results were statistically compared by a one-way ANOVA with multiple comparisons using Tukey's test (GraphPad version 7.0). Significant differences were considered at $p < 0.05$.

## 3. Results and Discussion

### 3.1. Brown A. bisporus Extraction

The fruiting body of brown *A. bisporus* mushroom and the physical appearance of brown *A. bisporus* extracts are shown in Figure 1. The physical appearance of the ethanolic extract (EE) was slightly different from the water extract (WE). Both extracts were a dark brown semisolid with the same characteristic odor. However, the color of the WE was darker than that of the EE. The percentage yields of both extracts are shown in Table 1. The results showed that the percentage yield of WE was higher than that of EE at $73.89 \pm 3.22\%$ and $2.64 \pm 1.39\%$, respectively. A previous study by Hu [19] reported that hot water extraction revealed a higher yield than ethanol extraction due to the breaking down of fungi cell walls so that high molecular weight compounds, such as polysaccharides, soluble carbohydrates, protein and free amino acids were obtained [20].

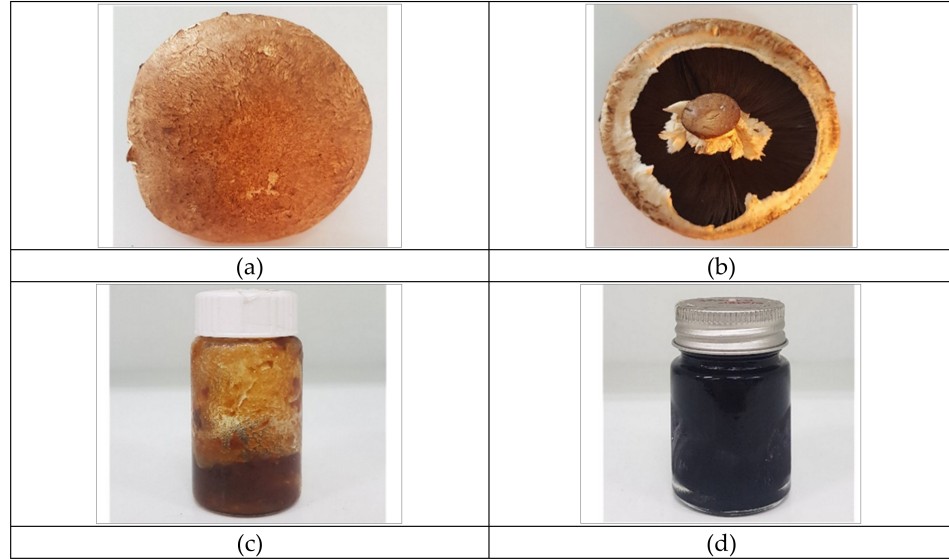

|     |     |
|:---:|:---:|
| (a) | (b) |
| (c) | (d) |

**Figure 1.** The physical appearances of (**a**,**b**) fruiting body of brown *A. bisporus*, (**c**) brown *A. bisporus* ethanol extract, and (**d**) brown *A. bisporus* water extract.

**Table 1.** The percentage of yields of brown *A. bisporus* ethanol extract (EE) and brown *A. bisporus* water extract (WE).

| Brown *A. bisporus* Extracts | Yield (%) |
|---|---|
| EE | $2.64 \pm 1.39$ |
| WE | $73.89 \pm 3.22$ |

*3.2. Measurement of Total Polysaccharides, Total Phenolics, and Total Flavonoids Contents*

3.2.1. Total Polysaccharides Content of Brown *A. bisporus* Extracts

The measurement of the total polysaccharides content with the anthrone method is based on the reaction of polysaccharides from the extracts, which is hydrolyzed into glucose by boiling in an acidic medium, leading to a green–blue color complex [21]. As is shown in Table 2, the results of total polysaccharides content of the extracts are expressed in mg of glucose per gram of extract. The total polysaccharides contents of the EE and the WE were $370.00 \pm 0.30$ and $734.00 \pm 0.03$ mg glucose/g extract, respectively. The polysaccharides content of WE was significantly higher than that of the EE ($p < 0.05$). Hot water extraction can extract water-soluble polysaccharides better than an organic solvent [22].

**Table 2.** The total polysaccharides, total phenolics, and total flavonoids contents of brown *A. bisporus* ethanol extract (EE) and brown *A. bisporus* water extract (WE).

| Brown *A. bisporus* Extracts | Total Polysaccharides Content (mg Glucose/g Extract) | Total Phenolics Content (mg GAE/g Extract) | Total Flavonoids Content (mg QE/g Extract) |
|---|---|---|---|
| EE | $370.00 \pm 0.30$ [a] | $184.87 \pm 0.10$ [a] | $16.11 \pm 0.21$ [a] |
| WE | $734.00 \pm 0.03$ [b] | $190.90 \pm 0.07$ [a] | $12.92 \pm 0.02$ [a] |

Different letters within a column indicate a statistically significant difference ($p < 0.05$).

3.2.2. Total Phenolics and Flavonoids Contents of Brown *A. bisporus* Extracts

Brown *A. bisporus* extracts were investigated for total phenolics and total flavonoids content. Firstly, total phenolics content was detected using the Folin–Ciocalteu method, which is based on electron transfer from phenolic compounds in the extracts to the reagent. The total phenolics contents in the EE and the WE were $184.87 \pm 0.10$ and $190.90 \pm 0.07$ mg GAE/g extract, respectively, as is shown in Table 2. The total phenolics content was not significantly different between both solvent extractions ($p > 0.05$). A previous study reported that the brown *A. bisporus* mushroom generally had higher phenolic compounds than other species, such as *Pleurotus ostreatus* and *Fomes fomentarius* [23].

The total flavonoids content was investigated using the aluminum chloride method, which is based on hydroxyl groups of flavonoids forming a yellow complex with aluminum chloride. The results are shown as mg QE/g extract as shown in Table 2. The flavonoids content of the EE was not significantly different from that of the WE ($p > 0.05$). The low level of flavonoids content might be due to the maturation stages of fruiting bodies. Obviously, both water and ethanol extracts of immature stage mushrooms showed higher phenolics and flavonoids compounds than mature stage mushrooms [9]. Previous studies have shown that phenolics and flavonoids compounds are found in edible mushroom extracts and have shown their ability to inhibit radical substances [24].

*3.3. Determination of Ergothioneine and Gallic Acid Contents of Brown A. bisporus Extracts*

The HPLC chromatograms of the brown *A. bisporus* extracts, gallic acid, and ergothioneine are shown in Figure 2. Gallic acid and ergothioneine can be detected at 280 nm at a retention time of 4.138 and 3.209 min, respectively. According to the HPLC results, ergothioneine and gallic acid were detected in the chromatogram of the EE at a retention time of 3.209 and 4.271 min, respectively. These compounds were also likely detected in the chromatogram of the WE at a retention time of 3.275 and 4.259 min, respectively. The ergothioneine and gallic acid contents of brown *A. bisporus* extracts are shown in Figure 3. The

ergothioneine level was significantly higher in the EE (357.79 ± 2.43 mg ergothioneine/g extract) than in the WE (184.78 ± 4.17 mg ergothioneine/g extract) ($p < 0.05$). Interestingly, ergothioneine has been reported to have antioxidant and anti-inflammatory activities [25]. The results indicated that the amount of gallic acid in the EE and the WE were significantly different at 24.69 ± 0.44 and 16.84 ± 0.23 mg gallic acid/g extract, respectively. *A. bisporus* is a good source of amino acids, such as alanine, glutamic acid, serine, and ergothioneine. It has been reported that the fruiting body of *A. bisporus* has high ergothioneine levels [8]. Phenolic compounds such as gallic acid, ferulic acid, and caffeic acid have been shown to have antioxidant, anti-aging, and anti-inflammatory effects. In addition, gallic acid was found to be the main phenolic acid in *A. bisporus* extracts in a previous study [26]. Therefore, from these results we can assume that ergothioneine and gallic acid are major abundant bioactive compounds in brown *A. bisporus* extracts that are related to the biological activities of the extracts.

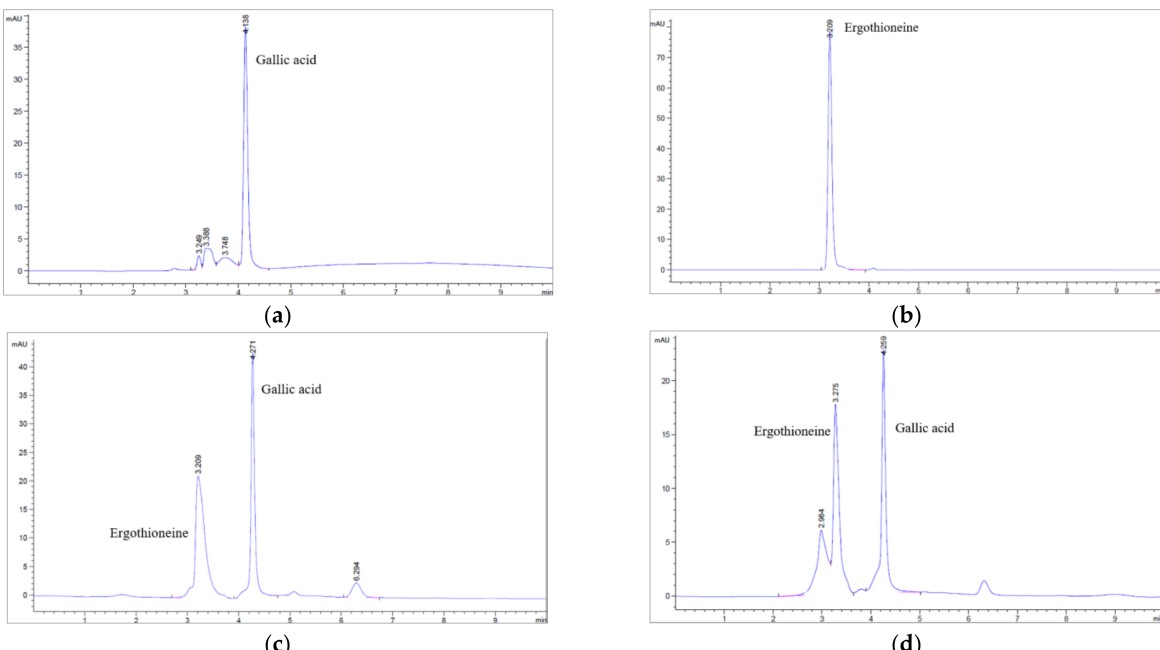

**Figure 2.** HPLC chromatograms of (**a**) gallic acid, (**b**) ergothioneine, (**c**) brown *A. bisporus* ethanol extract (EE), and (**d**) brown *A. bisporus* water extract (WE) using UV detection at a wavelength of 280 nm.

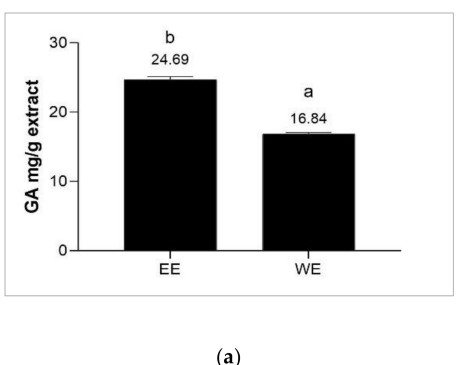
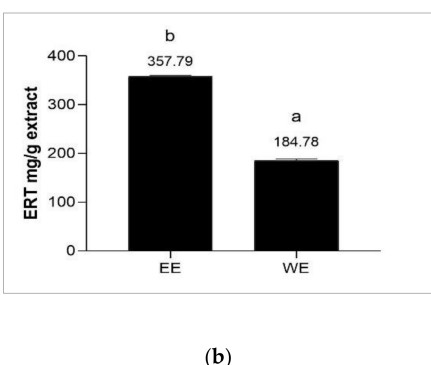

**Figure 3.** (**a**) Gallic acid (GA) and (**b**) Ergothioneine (ERT) contents of the brown *A. bisporus* ethanol extract (EE) and the brown *A. bisporus* water extract (WE). Different letters above the bars indicate statistically significant differences. Multiple comparisons of means are performed using Tukey's test at the 0.05 significance level.

### 3.4. Antioxidant Activity of Brown A. bisporus Extracts

Both extracts were evaluated for their antioxidant activity by DPPH, ABTS, and FRAP assays. Several assays of antioxidant activity are related to different mechanisms of antioxidants. DPPH and ABTS assays are based on electron transfer reactions that scavenge capacity by hydrogen donation. In contrast, the FRAP assay refers ferric iron ($Fe^{3+}$) complex to ferrous ion ($Fe^{2+}$) by electron donation as an ion reduction process [15]. The results are presented in Table 3. The EE possessed potent antioxidant activity in scavenging free radicals and reducing metal ability. Similarly, the EE possessed ABTS scavenging activity with a TEAC value of $8.06 \pm 0.08$ µM Trolox/g extract, which was not significantly different from that of the WE ($p > 0.05$). The antioxidant activity is correlated with the number of hydrogen atoms in phenolic compounds that donate to free radicals. Many phenolic compounds are obtained from the fruiting bodies of *A. bisporus* species such as gallic acid, trans-cinnamic acid, and p-coumaric acid [27]. In addition, ergothioneine exerts antioxidant properties with multiple mechanisms, including scavenging free radicals, chelating metal ions, and anti-inflammatory activity [28]. In a previous study, an ethanol *A. bisporus* extract possessed a strong antioxidant activity as determined by DPPH radicals, the FRAP method, and the bioactivity of phenolics, which was higher than that of other species according to the order *A. bisporus* > *P. ostreatus* > *P. eryngii* > *L. edodes* [29]. Due to the multiple mechanisms of antioxidant capacity, the extracts are an antioxidation biomarker that can reduce oxidative stress in the aging process. Therefore, the antioxidant activity of brown *A. bisporus* extracts can help to protect the skin barrier function from free radicals and an environment that has the potential for xerosis skin treatment.

**Table 3.** Antioxidant activity of brown *A. bisporus* extracts and standards by DPPH, ABTS, and FRAP assays.

| Brown *A. bisporus* Extracts | DPPH IC$_{50}$ (mg/mL) | ABTS TEAC Value (µM Trolox/g Extract) | FRAP FRAP Value (mg FeSO$_4$/g Extract) |
|---|---|---|---|
| EE | $0.30 \pm 0.04$ [b] | $8.06 \pm 0.08$ [a] | $390.50 \pm 0.32$ [b] |
| WE | $1.22 \pm 0.82$ [c] | $4.59 \pm 0.34$ [a] | $124.36 \pm 0.77$ [a] |
| Standards | | | |
| Ergothioneine | $0.01 \pm 0.05$ [a] | $125.64 \pm 0.69$ [b] | $646.36 \pm 0.06$ [d] |
| Gallic acid | $0.02 \pm 0.02$ [a] | $493.46 \pm 0.01$ [c] | $544.53 \pm 0.12$ [c] |
| Trolox | $0.02 \pm 0.08$ [a] | - | $481.81 \pm 0.20$ [b] |

Different letters within columns indicate statistically significant differences ($p < 0.05$). TEAC indicates the trolox equivalent antioxidant concentration as determined by ABTS assay.

### 3.5. Determination of Anti-Aging Activities

#### 3.5.1. Hyaluronidase Inhibitory Activity of Brown A. bisporus Extracts

In a turbidimetric method, hyaluronic acid is degraded by a hyaluronidase enzyme and turns into smaller fragments and monosaccharides [30]. The digested hyaluronic acid cannot bind with albumin and form a white precipitate. After the degradation, hyaluronic acid (undigested) with the hyaluronidase enzyme is detected via a reduction in turbidity [31]. The results of hyaluronidase inhibitory activity are shown in Figure 4. The EE and the WE can inhibit the hyaluronidase at $55.03 \pm 0.25$ and $55.94 \pm 3.98$%. Tannic acid inhibited hyaluronidase by up to $75.02 \pm 3.01$%, followed by ergothioneine and gallic acid. Brown *A. bisporus* extracts would be suitable for skin aging since they can inhibit the hyaluronidase enzyme by more than 50% as much as tannic acid. The hyaluronidase enzyme is the critical factor in hyaluronic acid turnover, which keeps skin moisturized and rejuvenated. Additionally, hyaluronic acid has high molecular hydrophilic properties and thus acts as a natural moisturizing factor and maintains skin moisture in the epidermis and dermis [32]. The disappearance of hyaluronic acid in the skin is observed during skin

aging, skin dryness, and skin barrier dysfunction. Therefore, the brown *A. bisporus* extracts, which can inhibit hyaluronidase, might be helpful for preventing skin aging and may be associated with the skin barrier healing potential alterations in xerosis.

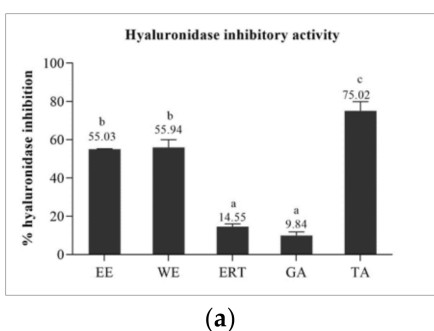

(a)

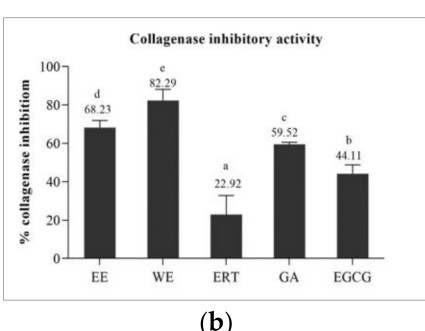

(b)

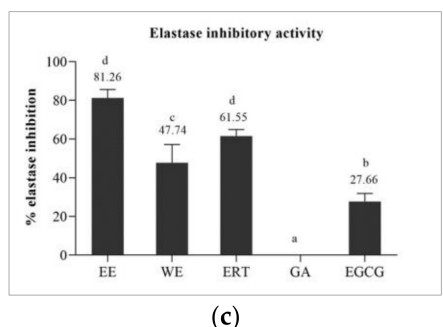

(c)

**Figure 4.** Inhibitory activities against (**a**) hyaluronidase, (**b**) collagenase, and (**c**) elastase of brown *A. bisporus* ethanol extract (EE), brown *A. bisporus* water extract (WE), ergothioneine (ERT), gallic acid (GA), tannic acid (TA), and epigallocatechin gallate (EGCG). Different letters above the bars indicate statistically significant differences. Different letters within columns indicate statistically significant differences ($p < 0.05$).

3.5.2. Collagenase Inhibitory and Elastase Inhibitory Activities of Brown *A. bisporus* Extracts

Collagen and elastin are essential proteins of the extracellular matrix in dermal connective tissue that maintains skin's strength and flexibility [33]. Collagen and elastin can be degraded by collagenase and elastase enzymes, leading to skin aging [34]. The inhibition effects of brown *A. bisporus* extracts on these two enzymes are shown in Figure 4. The EE and the WE can inhibit the collagenase enzyme at rates of $68.23 \pm 3.68\%$ and $82.29 \pm 5.89\%$, respectively, significantly higher than all standards ($p < 0.05$). From these results we can assume that the collagenase inhibitory activity of both extracts is related to ergothioneine and gallic acid contents. In addition, a previous study showed that ferulic acid and gentisic acid found in an *A. bisporus* extract also possessed collagenase inhibitory activity [35]. Moreover, the EE showed the highest inhibitory activity against the elastase enzyme at $81.26 \pm 4.37\%$, comparable to the WE ($47.74 \pm 9.45\%$). The explanation might be that the EE demonstrated high ergothioneine content, as was confirmed by a HPLC chromatogram. In this study, ergothioneine inhibited the elastase enzyme higher than gallic acid and epigallocatechin gallate ($p < 0.05$). These results are related to the study of Pientaweeratch [36], which reported that gallic acid showed weak activity on elastase inhibition. Many natural extracts present anti-collagenase and anti-elastase activities according to various phenolic compounds, such as gallic acid, caffeic acid, rutin, and ferulic acid [17]. From all the results mentioned above, the EE and the WE can inhibit two enzymes and delay the breakdown of collagen and elastin. Therefore, the extracts could be beneficial for xerosis treatment by reducing collagen and elastin degradation and improving firmness and hydration.

*3.6. Cell Culture*

3.6.1. Cytotoxicity Test by MTT Assay

MTT assay was used to measure the effect of brown *A. bisporus* extracts on the viability of HaCaT cells as shown in Figure 5. Cells were treated with various extract concentrations and the cell viability of the control sample was expressed as 100%. The results expressed cell viability of more than 80% with the WE at a concentration range of 37.5–300 μg/mL. In contrast, the EE only had an effect on cell viability when the concentration was 300 μg/mL. Hence, non-toxic concentrations of the EE and WE were chosen at 214.82 and 300 μg/mL for the anti-inflammatory activity study.

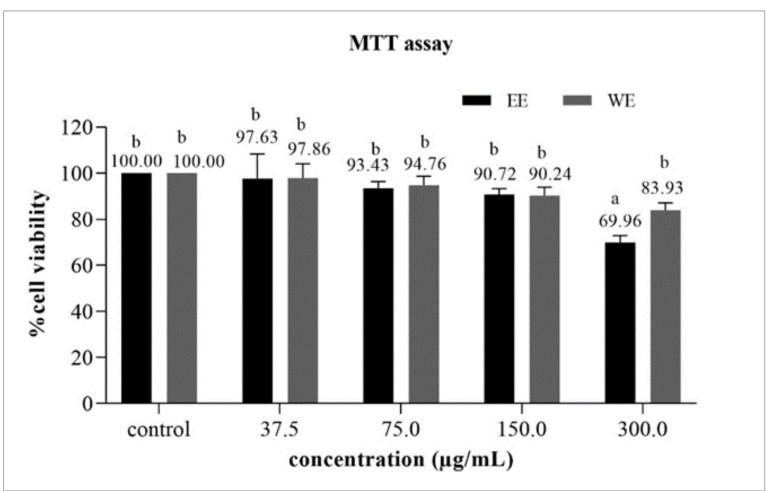

**Figure 5.** The cell viability of HaCaT cells treated with various concentrations of brown *A. bisporus* ethanol extract (EE) and brown *A. bisporus* water extract (WE) using MTT assay. Different letters above the columns in each extract indicate statistically significant differences ($p < 0.05$).

3.6.2. Preventive Effect of Brown *A. bisporus* Extracts against Inflammation in LPS-Induced HaCaT Cells

The secreted IL-6 and TNF-$\alpha$ on LPS-treated HaCaT cells were defined as 100%. The results of brown *A. bisporus* extracts were calculated as a percentage of secretion as shown in Figure 6. Inflammatory cytokines (IL-6 and TNF-$\alpha$) were activated with 1 mg/mL of LPS and incubated for 24 h. The percentage of LPS induced IL-6 secretion were significantly decreased by the EE (38.95 $\pm$ 0.46%) and the WE (38.00 $\pm$ 0.13%) pre-treatment as well as the extract alone ($p < 0.05$). Pre-treatment with EE reduced TNF-$\alpha$ secretion after induction with LPS, expressed as 29.55 $\pm$ 1.16%, as much as gallic acid. Pre-treatment with WE and ergothioneine can significantly decrease TNF-$\alpha$ levels ($p < 0.05$). EE and WE strongly inhibited cytokine production compared to ergothioneine and gallic acid alone. This might be due to other chemical compounds in the extracts that show synergistic effects with the interested compounds, thereby enhancing the bioactivity of the extracts. Therefore, it was demonstrated that pre-treatment of the WE and the EE showed a skin protection effect on HaCaT cells after being induced by LPS. The results were consistent with a previous report that suggested that ergothioneine has anti-inflammatory potential based on its ability to inhibit TNF-$\alpha$ expression [35]. It is well known that IL-6 and TNF-$\alpha$ are the main cytokines in many skin diseases that lead to skin inflammation produced by T-cells, macrophages, and keratinocytes [37]. IL-6 production is stimulated by histamine, which directly increases the itching sensation on the skin [38]. TNF-$\alpha$ stimulates the production of reactive oxygen species and various cytokines, including IL-6. In addition, high levels of IL-6 and TNF-$\alpha$ also induce skin inflammation and respond to abnormal epidermal permeability barriers, which likely reduces stratum corneum hydration [39]. The WE and the EE strongly prevented the secretion of IL-6 and TNF-$\alpha$ on HaCaT cells. Therefore, these findings suggest that brown *A. bisporus* extracts can be a natural ingredient to prevent skin conditions such as xerosis.

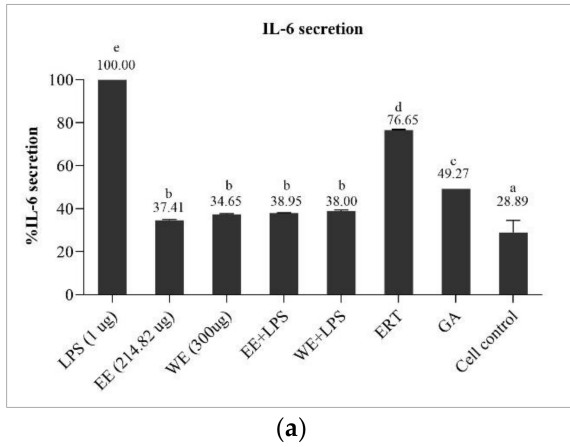

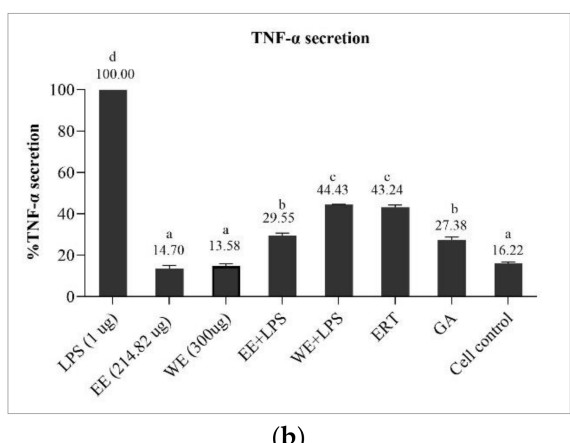

(**a**)                                                       (**b**)

**Figure 6.** The IL-6 (**a**) and TNF-$\alpha$ (**b**) secretion on HaCaT cells when treated with LPS alone, extract alone, pre-treatment with brown *A. bisporus* ethanol extract (EE), brown *A. bisporus* water extract (WE), ergothioneine (ERT), gallic acid (GA), and non-treated (cell control). Different letters above the bars indicate statistically significant differences ($p < 0.05$).

## 4. Conclusions

The present study demonstrated that the brown *A. bisporus* extracts contained amino acids such as ergothioneine and phenolic compounds such as gallic acid. The brown *A. bisporus* ethanol extract possessed the highest antioxidant activity when tested by the DPPH, ABTS, and FRAP assays. Moreover, it also exhibited inhibition against hyaluronidase and elastase enzymes, whereas the WE showed inhibitory activity against the collagenase enzyme. None of the extracts exhibited cytotoxicity and were able to inhibit IL-6 and TNF-$\alpha$ secretion on HaCaT cells. Thus, the beneficial effect of the brown *A. bisporus* extracts can be used as a natural ingredient to reduce skin aging and skin inflammation, associated with healing the skin barrier function for xerosis treatment.

**Author Contributions:** Conceptualization, K.K.; methodology, K.K., P.L., O.N., N.I. and N.N.; software, N.N. and K.K.; validation, K.K., N.N. and N.I.; formal analysis, N.N., K.K. and N.I.; investigation, K.K., N.N. and N.I.; resources, K.K. and N.I.; data curation, K.K. and N.N.; writing—original draft preparation, N.N. and K.K.; writing—review and editing, K.K., P.L., O.N., N.I. and N.N.; visualization, N.N. and K.K.; supervision, K.K.; project administration, K.K.; funding acquisition, K.K. All authors have read and agreed to the published version of the manuscript.

**Funding:** This research project was partially supported by Chiang Mai University, Faculty of Pharmacy, Chiang Mai University and Teaching Assistant and Research Assistant Scholarships (TA/RA) academic year 2020 from Chiang Mai University. In addition, this research project was financially supported by the Agricultural Research Development Agency (Public Organization) of Thailand (ARDA).

**Institutional Review Board Statement:** Not applicable.

**Informed Consent Statement:** Not applicable.

**Data Availability Statement:** The data presented in this study are available within the article.

**Acknowledgments:** The authors would like to acknowledge Chiang Mai University and the ARDA of Thailand for their financial support. The authors would also like to acknowledge the Faculty of Pharmacy, Chiang Mai University for the research grant and facilities used in the project. In addition, the authors would also like to acknowledge the Division of Clinical Microscopy, Department of Medical Technology, Faculty of Associated Medical Sciences, Chiang Mai University, Chiang Mai, Thailand for the facilities used in the project.

**Conflicts of Interest:** The authors declare no conflict of interest.

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
