# Peer review of "Potential and Alternative Bioactive Compounds from Brown Agaricus bisporus Mushroom Extracts for Xerosis Treatment"

_scipharm, doi:10.3390/scipharm90040059_

Round 1

Reviewer 1 Report

Dear Authors,

I write you in regard to the manuscript "Potential and Alternative Bioactive Compounds from Brown Agaricus bisporus Mushroom Extracts for Xerosis Treatment".

- please, Abstract must be revised and rewritten. 

- in lines 129, 142 and 155, substitute calibration equation with analytical curve.

- test 2.9 must be revised. How were the samples maintained? 

- results and discussion will have to be revised after adding new details to the test.

- conclusions must be revised. Safety of the samples can not be a conclusion, since the selected test did not lead to such end.

- the text speculated a possible alternative for the treatment for xerosis, therefore, this data should be removed from the title and from conclusions.

Author Response

Dear reviewer:

We greatly appreciate the reviewers' valuable comments and suggestions. We have carefully read and responded to all comments, point by point in the attachment. The specific alterations in the manuscript in response to reviewer comments are shown in Task change.

We hope all of the changes have addressed the reviewers' concerns, and we hope our manuscript will be accepted for publication.

Sincerely,

Asst.Prof.Dr. Kanokwan Kiattisin

Reviewer 2 Report

1. The manuscript is well written with interesting results. However, authors need to discuss more on the bioactivities of the extract as compared to previous published data.

2. Line 33: The unwelcome visible signs are skin sensitivity, inflammation, and dryness. This sentence is incomplete.

3. In section 2.5, authors stated that the extract of A.bisporus was rich in amino acid and phenolic compounds, but no literature was cited to support this claim.

4. In line 367-368: However, the results indicated that the amount of gallic acid in the EE and the WE were not significantly different at 24.69 ± 0.44 and 16.84 ± 0.23 mg gallic acid/g extract, respectively. What kind of statistical analysis was used to compare these data? Since we can see clearly by looking at the standard deviation value that amount of gallic acid in EE was significantly higher than WE. Please separate the statistical analysis of gallic acid to ERT, so that the result is not bias (Figure 3)

5. In section 3.4, the authors presented data of the antioxidant activity of two extracts, which indicated a quite strong antioxidant activity of EE. To confirm this, please provide data which comparing the present data to previous published data of A.bisporus extract-mediated antioxidant activity in vitro. This is important to specify that the extraction method, the mushroom specimen used in this study can be regarded as reliable procedures, compared to previously used methods.

6. Similar response as no.4 is addressed to the MTT assay, total polysaccharides and flavonoid compounds. Why did the author measure the polysaccharides content on the extract? What correlation does the polysaccharide content data have towards antiaging or antioxidant activities?

7. Line 455-457, Cells were treated with various extract concentrations and the cell viability of the control sample was expressed as 100%. The result expressed cell viability more than 80% without affecting cellular morphology at the WE at a concentration range of 37.5 - 300 µg/mL. What kind of cellular morphological change were observed in this assay? Such statement was not supported by data.

8. In figure 6, authors argue that EE and WE which were rich in ERT and GA could reduce the secretion of inflammatory-marker cytokines, however we can clearly observed that addition of ERT and GA alone increased the cytokine production compared to that EE and WE. These data indicate that actually bot ERT and GA plays the major role in anti inflammatory activity, other compounds might elicit such activity, instead. Thus, it is important to discuss such phenomenon.

9. Authors used 1 mg/ml extract for the in vitro assay of antiaging activity, yet as maximum as 300 ug/ml extract was used for cytotoxicity assay. Thus, it is necessary to confirm that such high 1 mg/ml extract concentration which elicit antiaging activity may not be toxic to cells as well. In addition it would be more informative to express the cytotoxicity assay data as LC50 (use positive control for compariso) rather than using various concentrations range as presented on figure 5.

10. In conclusion section, authors stated the safety of the extract since as based on MTT assay, however it is unlikely that since at least LC 50 value was not available.

Author Response

Dear reviewer:

We greatly appreciate the reviewers' valuable comments and suggestions. We have carefully read and responded to all comments, point by point in the attachment. The specific alterations in the manuscript in response to reviewer comments are shown in Task change.

We hope all of the changes have addressed the reviewers' concerns, and we hope our manuscript will be accepted for publication.

Sincerely,
Asst.Prof.Dr. Kanokwan Kiattisin
Corresponding author

Round 2

Reviewer 1 Report

Dear Authors,

I write you in regard to the new version of the manuscript "Potential and Alternative Bioactive Compounds from Brown Agaricus bisporus Mushroom Extracts for Xerosis Treatment".

- in 2.9, a model membrane was used. It was confusing to comprehend the measurement of vital properties from skin removed from an experimental animal. There is no guarantee that superficial hydration and TEWL would represent the in vivo model.

Author Response

Dear reviewer:

We greatly appreciate the reviewers' valuable suggestions and concerns. We have carefully read and responded to all comments, point by point in the attachment. The specific alterations in the manuscript in response to reviewer comments are shown in Task change and highlight.

We hope all of the changes have addressed the reviewers' concerns, and we hope our manuscript will be accepted for publication.

Sincerely,
Asst.Prof.Dr. Kanokwan Kiattisin
Corresponding author

Reviewer 2 Report

Authors have responded the questions properly. Just text editing is necessary to avoid misstyping on the manuscript.

Author Response

(The authors gave the same response as above.)

Round 3

Reviewer 1 Report

Dear Authors,

I thank you for the manuscript improvement. Unfortunately, the use of the pig skin, as presented, would not be suitable for the tests without maintaining the dynamic of water on the tissue, for example, using a difusionismos cell. consider to suppress that part.

Author Response

Dear reviewer:
We greatly appreciate the reviewers' valuable suggestions and concerns. We have carefully read and responded to the comments. The specific alterations in the manuscript in response to reviewer comments are shown in Task change.
We hope all of the changes have addressed the reviewers' concerns, and our manuscript will be accepted for publication.
Sincerely,
Asst.Prof.Dr. Kanokwan Kiattisin
Corresponding author
